# In-Line Holography in Transmission Electron Microscopy for the Atomic Resolution Imaging of Single Particle of Radiation-Sensitive Matter

**DOI:** 10.3390/ma13061413

**Published:** 2020-03-20

**Authors:** Elvio Carlino

**Affiliations:** Istituto per la Microelettronica ed i Microsistemi, Consiglio Nazionale delle Ricerche (CNR-IMM), Sezione di Lecce, Campus Universitario, via per Monteroni, 73100 Lecce, Italy; elvio.carlino@cnr.it

**Keywords:** TEM, in-line holography, single particle imaging, atomic resolution imaging, radiation damage, soft matter, nanostructured drugs, organic materials

## Abstract

In this paper, for the first time it is shown how in-line holography in Transmission Electron Microscopy (TEM) enables the study of radiation-sensitive nanoparticles of organic and inorganic materials providing high-contrast holograms of single nanoparticles, while illuminating specimens with a density of current as low as 1–2 e^−^Å^−2^s^−1^. This provides a powerful method for true single-particle atomic resolution imaging and opens up new perspectives for the study of soft matter in biology and materials science. The approach is not limited to a particular class of TEM specimens, such as homogenous samples or samples specially designed for a particular TEM experiment, but has better application in the study of those specimens with differences in shape, chemical composition, crystallography, and orientation, which cannot be currently addressed at atomic resolution.

## 1. Introduction

Transmission Electron Microscopy (TEM) is widely used to study the properties of matter at the highest spatial resolution. There is a wide body of literature that reports on the study of single nanoparticles of inorganic material, showing how fundamental subtle physical effects can be understood by TEM experiments at atomic resolution [1]. High-Resolution TEM (HRTEM) enables direct access to the structural properties of individual particles, correlating their structure to their behavior. This allows the comprehension of matter at the atomic level and the development of new materials for a huge variety of applications [2]. In a realistic specimen, the particles are not necessarily all equal and properly oriented, but they could have different crystalline properties, defects, or allotropic state, which deeply influence the properties of the materials system. The study of single particles enables, in a batch of nanoparticles, the distinguishing of the differences between the particles and the relevant influence on the macroscopic behavior of the nanostructured material. In the case of radiation-sensitive material, standard HRTEM approaches on single particles could fail, due to the damage induced by the high-energy electrons on the specimen. This is a big issue for biologic materials [3]. Moreover, in pharmacy, TEM study of single nanocrystalline salt drugs, which are organic-like material consisting of low atomic number atoms tied with weak chemical bonds, cannot be performed by standard HRTEM [4,5]. In fact, radiation damage provides the main limitation to the spatial resolution of electron beam imaging or spectroscopy of organic materials [6]. The use of new TEM/STEM microscopes, equipped with aberration correctors and field emission cathodes, ascertains the highest spatial resolution so far achieved, but this technology can deliver a high-density of current on the specimen, making radiation damage an issue of growing importance also for inorganic materials and even metals [7]. As the radiation damage cannot be eliminated, there is a strong interest in finding approaches that can limit, or in some way overcome, the damage, enabling the study of the specimen at atomic resolution despite the radiation damage [8,9]. The issue of radiation damage is not only limited to electrons, but it is common to other probes, for example, X-rays. A very recent approach, which is polarizing the X-ray scientific community, is the so-called diffract-and-destroy method, developed for extremely intense sources of X-rays with high-frequency pulses, such as free electron lasers (FEL) [10]. In this case, the scattered photons are acquired on a femtosecond time-scale, and therefore before the explosion of the illuminated ensemble of molecules. Unfortunately, this approach cannot be straightforwardly extended to the case of electrons in a TEM, due to the peculiarity of electrons as charged particles and the relevant Boersch effect [9,11]. The radiation damage depends strongly on the specimen nature. Indeed, in a very schematic description, the knock-on damage is the main issue in conducting materials, whereas the ionization damage, and the related radiolysis, is the main issue in case of semiconducting or insulating inorganic materials and for all organic or organic-like matter. The knock-on damage is due to the deflection of the primary electrons by the electrostatic field of the nuclei of the specimen. For deflection angles up to 100 mrad, this scattering is considered elastic, to a good approximation, as the energy transferred to the specimen nucleus is below 0.1 eV. For angles of deflection higher than 100 mrad, this event can cause an energy transfer from the incident electron to the nuclei of the specimen of several eV, producing sputtering of the atoms from the specimen surface. For energy transfer of tens of eV, the atoms are displaced in the specimen, forming crystal defects [12]. In the case of ionization damage, known as radiolysis, the scattering considered is between electrons: the primary electrons lose part of their kinetic energy by ionizing the atoms of the specimen or exciting collective motion in the form of plasmons, which involves, in the case of metals, the vibration of atomic nuclei, but not their permanent displacements [6]. In the case of insulating or semi-insulating materials, the energy loss by the primary electrons produces holes that are not rapidly recombined, as they are in metals, but could result in a stable arrangement that stores the energy loss by the primary beam in a configuration of broken bonds [6]. Energy of few eV is enough to break a chemical bond, whereas the ionization energy involves tens of eV, and most of this energy is dissipated by producing secondary electrons. Consequently, an electron made available from an ionization process during its lifetime can break several bonds in an organic material, producing most of the damage in this class of materials [6]. The breaking of the bonds produces the loss of short-range order in crystalline materials and results in the appearance of a diffused halo in the diffraction pattern. It is a common and frustrating experience, during electron diffraction experiments in a TEM on radiation-sensitive materials that, as a function of the irradiation time and dose rate, the diffracted Bragg’s spots are quickly progressively faded and disappear completely, as a result of the disruption of the crystalline order. An example is shown in Figure 1.

Figure 1 shows two selected area electron diffraction patterns obtained by illuminating an area of ~10 microns in diameter of a specimen consisting of nanoparticles of a salt of Vincamine [13], which is an indole alkaloid used for the treatment of important neurovegetative conditions, like Parkinson’s and Alzheimer’s diseases [14,15]. The current density of the electron probe is ~300 e^−^Å^−2^s^−1^. In Figure 1a, the specimen has been exposed to the electrons for 0.01 sec, whereas in Figure 1b, the same area of the specimen is exposed to the electron beam for ~0.3 sec, which faded or cancelled the diffracted intensities, due to the damage of the crystalline structure. As reported by Egerton [6], the radiation damage is of particular concern in electron microscopy because of the need of high spatial resolution. Otherwise, we could simply defocus the incident beam and spread the damage over a large volume of the specimen; the fraction of the broken bonds would then become small and the radiation damage would cease to be a problem. It is therefore rather evident that imaging of single particle, in a specimen like the one shown in Figure 1, is quite cumbersome. Imaging can become impossible if we aim to look for a specific particle of interest before acquiring the atomic resolution HRTEM image. Indeed, the limit imposed by the radiation damage is particularly evident in the case of atomic resolution TEM study of nanoclusters, where when the beam is focused onto a beam sensitive particle, it can rapidly cause the disappearance of the crystalline order and, furthermore, the effect of charging could even produce a Coulomb explosion creating a hole in the eventual supporting film used for TEM observation. Therefore, especially in the case of a specimen consisting of nanoparticles, where the interest is to study the properties of single particles, before acquiring an atomic resolution HRTEM image, we need to find the particle suitable for the experiment. A suitable particle is defined as a particle properly oriented with respect to the electron beam to distinguish between eventual different crystal polytypes and orientation. The HRTEM image contrast is due mainly to the elastic scattering between the electrons and the nuclei of the atoms of the specimen [2]. Unfortunately, organic particles consist of low atomic number atoms, which have a relatively low elastic scattering power [2], and therefore a relatively high density of current is necessary to distinguish the particle with respect to the supporting film. Nevertheless, once the candidate particle is found, we need to evaluate the relevant diffraction pattern to check the proper orientation, therefore we need to wait until the eventual specimen holder drift is stopped and only at this point, after focusing, we can acquire the HRTEM image. Indeed, in the case of radiation-sensitive materials, we cannot even detect the particle by standard specimen survey conditions without destroying the particle itself or its eventual crystalline order. For example, in the case of radiation-sensitive polymers, to avoid quick degradation of the material, the electron current density threshold should be between 0.1 to 10 e^−^Å^−2^s^−1^ [16]. In these conditions, if we are looking for a particle in a TEM specimen with a low density of nanoparticles, we do not have enough image contrast to distinguish between the nanoparticle itself and the supporting carbon film. It is, therefore, important for a successful atomic resolution low-dose HRTEM experiment on radiation-sensitive specimen, to find an approach that enables the detection of an isolated particle, to check its crystallinity and orientation with respect to the electron beam, to check the specimen holder drift to enable atomic resolution imaging, and to adjust the electron optical conditions prior to low dose and low dose rate HRTEM acquisition, all by using an electron density of current between 0.1 to 10 e^−^Å^−2^s^−1^. The use of expediencies to reduce the radiation damage in some classes of materials can be applied; this is the case, for example, in the use of a conducting coating on inorganic insulators to reduce the effect of radiolysis, the use of energy primary beam as low as 60 KeV to avoid the knock-on damage in many materials, or the use of low dose rate to enable the specimen recovery of knock-on damage in some classes of materials. There exists extensive literature on where these expediencies are considered and their effectiveness is discussed [6,9,17,18,19,20]. In organic materials and soft matter, a particular role is played by the use of low temperature to reduce the effect of radiolysis. In fact, for a long time, it has been recognized that the use of cryogenic temperature reduces the appearance of the effect of radiolysis in TEM images of proteins [21,22]. Nevertheless, the use of cryogenic temperature still needs the use of an extremely low dose of electrons, and this results in TEM images with extremely low contrast, where in some cases, the presence of a single particle in an electron micrograph can be hardly distinguished [21]. Actually, the effect of cryogenic temperature does not influence the ionization cross section, but reduces the desorption and the movement of the molecular fragments, whose bonds have been broken by the primary and secondary electrons. For example, at liquid nitrogen temperature, the density of electrons to image quite safely a protein has to be ≤5 e^−^Å^−2^, whereas at the liquid helium temperature, it has to be ≤ 20 e^−^Å^−2^ [21]. This is why nowadays the approach, which is revolutionizing the structural biology in TEM, is the cryo-EM, which is a method that enables the study of proteins that cannot be easily crystallized for study by X-ray crystallography, for example, the membrane proteins. The word “cryo” denotes that this method is performed at cryogenic temperature. The development of this approach earned the authors the 2017 Nobel Prize in Chemistry [23]. Cryo-EM was actually established as method decades ago and requires the acquisition of thousands of low dose images from a specimen consisting of identical particles. The images are then processed by sophisticated software that produces a tridimensional model of the particle. Note that cryo-EM is not a true imaging approach, and the resulting structural model needs to be validated by well-controlled protocols, whose development is still in progress with dedicated task forces. The validation of structural models is a field where there is a strong research activity, and the number of structures solved by cryo-EM is growing fast [24]. The biggest success in the last years of cryo-EM is due to some particular technological advances that represent a turning point in the results achievable by cryo-EM, producing the so-called “Resolution Revolution” [25]. One key technological advance for the practical use of cryo-EM has been the development of direct conversion detectors to acquire the electron images with much better performances for the same low dose of electrons [26,27]. The other key point has been the development of capacity of calculus, which was hard to conceive when the basics of cryo-EM were proposed. These aspects were underlined by Richard Henderson during his Nobel Prize acceptance speech. It is worthwhile to remark once again, that the result of a so-called single-particle imaging by cryo-EM, is not a true single particle imaging at all, as it uses the images of thousands of particles, assumed as identical, to produce a 3-D model of the macromolecule. This reconstruction is, in any case, achieved at a spatial resolution worse than the one allowed by the electron optics, and related to the statistics of the particles imaged and also to the accuracy of the starting model of the particle to be imaged.

Here, we report on the use of in-line holography in TEM to perform true single-particle atomic resolution imaging of soft matter and biologic nanoparticles, believed not accessible by high-energy atomic resolution TEM experiments. In the reported studies, the experiments were successfully performed at room temperature using 200 KeV electrons. In-line holography is used to survey the specimen, to find suitable isolated particles, and to tune the experimental conditions to enable a reliable quantitative atomic resolution imaging experiment on radiation-sensitive materials. This approach enables to acquire safely low dose and low dose rate HRTEM images from radiation-sensitive materials, gathering information in analogy with the well-established methodology used in materials science on specimens robust to electron irradiation.

## 2. Methods

The imaging and analysis performances of a TEM, reported by the manufactures, have meaning only if the numbers of electrons “*N*”, measured within each spatial resolution element, have a statistical significance. The radiation damage limits the number of electrons for each resolution element of size *δ* and, therefore, is directly correlated to the resolution [28,29]. If *D_c_* is the critical electron dose that the specimen region δ can tolerate without damage then
(1)$$N=F\left(\frac{D_{c}}{e}\right)\delta^{2}$$
where “*e*” is the electron charge and *F* is the fraction of electrons reaching the detector. The image contrast C, between the recorded pixel and its neighbors containing N_b_ electrons, is defined as
(2)$$C=\left(N\;–\;N_{b}\right)/N_{b}=\Delta N/N_{b}$$

(C is negative in the case of absorption contrast).

The signal to noise ratio (SNR) is:(3)$$\left(\mathrm{SNR}\right)=\sqrt{\left(DQE\right)}\left(\Delta N/N_{shot}\right)$$
where DQE is the detective quantum efficiency, which is a measure of the noise introduced by the detector, and $N_{shot}=\sqrt{N+N_{b}}$, according to Poisson’s statistics. 

Therefore, from Equations (1)–(3), the size of δ depends linearly on the SNR. The dose-limited resolution is thus
(4)$$\delta =\frac{\left(SNR\right)}{\sqrt{\left(DQE\right)}}\frac{\sqrt{2}}{\left|C\right|}\frac{1}{\sqrt{FD_{c}/e}}$$

Equation (4) gives analytical evidence, in the approximation of weak contrast [30], of the role of the radiation damage on the resolution. It is worthwhile to mark the role of *F*, which makes bright-field imaging intrinsically less efficient than phase contrast in terms of radiation damage limited resolution [29], irrespective to the peculiar features of the two imaging methods. Moreover, the role of DQE should be noted, as it is the reason why, in the last years after the introduction of direct conversion detectors with better DQE, we are observing a fast growth of microscopes equipped with direct conversion devices especially on the instruments dedicated to cryo-EM, but not limited to them. The Rose’s criterion states that, to distinguish two adjacent elements in an image, the SNR has to be at least 5 [31]. From Equation (4) and considering *F* = 1, for HRTEM, and a perfect detector, resulting in DQE = 1
(5)$$\delta =\left(5\frac{\sqrt{2}}{\left|C\right|}\right)\sqrt{\frac{e}{D_{c}}}$$

According to this equation, we can estimate the resolution limit for a polymer with a reasonable *D_c_* of 0.01 C/cm^2^. Considering a contrast of 20% (C = 0.2) it results δ~1 nm [32]. The high sensitivity of some classes of materials, like polymers, nanodrugs, or biologic matter, limits the resolution in an imaging experiment, due to the need to use low dose of electrons. Furthermore, these materials have a relatively small cross section for elastic scattering with electrons resulting in a poor image contrast. This makes it impossible to perform an experiment of atomic resolution imaging on single radiation-sensitive nanoparticle, as its intrinsic low scattering power, together with the fast damage, limits the particle visibility and the tuning of the electron optical conditions for quantitative imaging before its damage. An example is shown in Figure 1. Indeed, by using standard survey methods, like bright-field or HRTEM, to find the particles suitable for quantitative atomic resolution imaging experiments, we are groping in the dark looking for where the representative nanoparticles are. Highly defocused electron probe coupled to strong defocus of the objective lens, or eventual future dedicated phase plates with improved stability with respect to charging effect and contaminations [33] could help to increase the contrast in the sample survey to detect the nanoparticles. Nevertheless, once the particle has been seen, we still do not have any clue of its crystallinity and orientation, and in the time necessary to put in focus the objective lens and to stabilize the eventual drift for an atomic resolution image, the particle would have been destroyed or, at least, it would be no longer representative of the pristine particle. The use of an in-line hologram could overcome these difficulties as it provides high contrast evidence of the nanoparticle, even if it is composed of light chemical elements, while providing clues on its crystalline status and on the electron optical conditions. All this required using density of electron current of <1 e^−^Å^−2^s^−1^. In-line electron holography was proposed in 1948 by Dennis Gabor for overcoming the limitation related to electron lens aberrations [34], and his idea was awarded in 1971 by the Nobel prize in physics. The in-line hologram formation is schematized in Figure 2.

Electron holography in TEM historically followed, for its application in physics, biology, and materials science, mostly another experimental configuration called off-axis holography, which requires a reference wave field obtained by splitting the illuminating wave field before the interaction with the specimen by using, in most of the cases, an electron biprism [35]. The reference is then used to recover the wave field after the scattering with the specimen. The introduction of the off-axis holography is related to practical reasons for quantitative applications of the holography to overcome the “twin image problem” related to the original proposal of Gabor [36,37,38,39]. Although for many years after its proposal in-line holography was not used for quantitative imaging, this experimental configuration is well known, for example, to all those electron microscopists involved in convergent beam electron diffraction (CBED) or scanning transmission electron microscopy (STEM). Indeed, this approach was used, for example, as a visual aid in CBED to follow the movement of the specimen in the direct space (shadow image) while observing the diffraction pattern in the reciprocal space during tilting for accurate specimen orientation with respect to the electron beam [40]; or in STEM, to accurately tune the microscope illumination lenses alignment necessary for accurate quantitative experiments. In-line holograms are also used for lens aberrations measurements and corrections [41], for accurate focusing [42], and for a variety of applications, as recognized in the pioneering work of J. M. Cowley [43]. In fact, as far as the detection of a low scattering nanoparticle concerns, the presence of the twin-image effect is an advantage as it enhances the contrast of the particle in the hologram. This feature of the in-line hologram, is here used to set up the method for the very low dose survey and to tune the electron optical set up to enable HRTEM image on single particle. It is indeed worthwhile to mark here that, recently, thanks to the advances in digital holography, and in particular in phase-shifting digital holography [44], the “twin image problem” has been successfully overcome, enabling an in-line hologram reconstruction free from twin-image disturbance, and therefore making feasible the retrieval of the relevant intensity and phase distribution [45], establishing in-line holography itself as a possible quantitative approach to atomic resolution imaging.

The experimental conditions in the in-line holography, and in particular the electron current density in the area of interest, can be readily changed by simply acting on the microscope illumination conditions, and therefore in-line holograms can be formed and observed in the reciprocal space varying the density of the electrons on the specimen to extremely low value, but still effective to detect the shadow image of the particles and, at the same time, looking at diffraction coming from the illuminated area, as shown in Figure 3. When the electron probe is focused above or below the specimen plane, each diffracted disc in the reciprocal plane contains a shadow image of the direct plane of the specimen. The magnification “M” of this shadow image is related to distance “u” between the focal plane and the specimen, and to the distance “v” between the specimen plane and the plane of view. From the geometric optics, M = v/u. The magnification of the image can be readily changed by changing the plane where the electron probe is focused [46].

The hologram in Figure 3 comes from an area of ~150 nm in diameter, illuminated by an electron density of ~0.1 e^−^Å^−2^. The specimen consists of vinpocetine and polyvinylpyrrolidone (PVP), and was studied at room temperature as no cooling holder was necessary, despite the extreme sensitivity of the material to the radiation damage. The electron energy was 200 KeV. The low dose of electrons enables surveying of the sample finding an area suitable for atomic resolution imaging with no detectable degradation of the particles. The size of the circular area in the diffractogram is related to the size of the used condenser aperture, whereas the magnification of the particles and their contrast are related to the illumination condition and in particular to the spatial coherence of the probe [47]. In the circular area in the central part of Figure 3, the presence of spherical particles of vinpocetine and filaments of polyvinylpyrrolidone is clearly detectable in the shadow image, despite the low intensity of the electron probe and the low atomic number of the atoms in both vinpocetine and PVP [48,49]. The shadow image also enables the accurate monitoring of the eventual drift of the specimen and checking when the conditions are such that an HRTEM from the particle of interest can be safely recorded with a low dose rate. In the dark area around the region selected by the condenser aperture, the sharp diffracted intensities are the signature of the crystalline nature of some of the structures in the illuminated area. The advantage of operating in the reciprocal space is evident, as the hologram contains all the information on the shape and crystal status of the particle of interest and also on the electron optical conditions, whereas the focus of the objective lens, adjusted in the direct space on an area close to the area of interest before switching the electron optics to diffraction mode, remains fixed and it is recovered immediately by switching back the electron optics to conjugate the direct space to the detector. This is the optimal condition to acquiring the relevant low dose and low dose rate HRTEM image, as shown in Figure 4.

Indeed, in Figure 4, the HRTEM image of a particle of vinpocetine together with the relevant diffractogram is shown. The density of electrons to image the particle is ~100 e^−^Å^−2^. The lattice contrast in the particle is rather sharp indicating a high degree of crystal order, as also confirmed by the relevant diffractogram. Note that the experiment was performed at room temperature by using a JEOL 2010 FEG UHR TEM/STEM (Jeol ltd., Tokyo, Japan) operated at 200 kV. The FEG cathode enables to illuminate the specimen by a highly coherent probe of electrons, whereas the Cs = (0.47 ± 0.01) mm of the objective lens provides an interpretable spatial resolution at optimum defocus for HRTEM of 0.19 nm [2]. The equipment was operated in free lens control to finely tune the illumination conditions [4,5]. The beam current was measured by Faraday’s cup. This equipment and the above reported experimental conditions were used for a variety of successful experiments on radiation-sensitive materials, and some of these experiments are discussed in the next paragraph.

## 3. Results and Discussion

The method for atomic resolution imaging of radiation-sensitive materials by in-line holography coupled to HRTEM has been extensively applied, in our laboratory, to drug salts, but also to biologic samples, enabling the achieving of atomic resolution imaging despite the use of primary electrons of 200 KeV and specimen at room temperature [4,5]. The results shown in this paragraph focus on the low dose and low dose rate HRTEM experiments, giving significant atomic resolution insights of the nanoparticles of radiation-sensitive matter, however note that the HRTEM experiments follow the in-line holography survey of the specimen and the procedures, described in the methods section, necessary for accurate survey, electron optical tuning, and low-dose HRTEM experiments at low dose rate.

In the following, the results on vincamine nanoparticles (Section 3.1), co-crystals nanoparticles of caffeine and glutaric acid (Section 3.2), and nanoparticles of ferrihydrite bound to creatinine particles (Section 3.3) are reported.

### 3.1. Vincamine

Here, the in-line holography method was applied to the study of the nanoparticles of vincamine citrate as obtained by “solid-excipient assisted mechanochemical salification”. The aim was to correlate the enhancement of solubilization kinetics of ball-milled vincamine citrate with respect to the vincamine citrate obtained by classical synthetic routes and, more generally, to understand the structural origin of the different features of this drug obtained by different methods of synthesis [4]. The analysis of single particles repeated on hundreds of particles, within the limit of a statistical analysis by HRTEM experiments, enables acquiring information on the different crystallographic properties related to the differences in the material preparation. Moreover, this leads to a better understanding of the results of the X-ray diffraction pattern measurements, as far as the peak broadening is concerned, and photoelectron spectroscopy experiments performed on the same specimens [4]. The HRTEM images in Figure 5 and Figure 6, with the relevant diffractograms, were obtained by exposing the particles to a parallel electron beam and illuminating a relatively large area, of ~100 nm in diameter, of the specimen around the particle of interest. The density of electrons in the illuminated areas was of ~100 e^−^Å^−2^ and no evidence of electron-induced damage was detected. It is worthwhile to remark that during these kinds of experiments, we noticed the role of the dose rate on the particle damage, namely, low dose rate has relatively little effect on the particles damage [50], at the same total dose delivered, as the structure has the time to recover the damage between different collisions events and in images series at low dose rate the particles can maintain their pristine structure [51,52]. During the in-line holography survey with low density of electron current, typically between 0.1 and 10 e^−^Å^−2^s^−1^, the features of the diffracted spots remained unchanged for tens of seconds but, if the density of electron current is increased, by changing the excitation of the condenser lens in a range of 10^2^ to 10^3^ e^−^Å^−2^s^−1^, the diffracted spots become immediately faded and disappear. Figure 5 shows a particle of vincamine oriented along a high symmetry zone axis, in the relevant diffractogram the spots correspond to a lattice spacing of (0.20 ± 0.01) nm, which is a typical value for many polytypes of vincamine. The experiments show that about 10% of the observed particles have a crystalline nature, whereas in the remaining cases, the particles are amorphous. In the case of the crystalline particles, the availability of measurable symmetries in the diffractogram enables not only to measure the lattice spacing but also to compare the experimental symmetries with those simulated by using the known allotropic states of Vincamine. This results in a higher accuracy in the relative measurement of spacing in the same diffractogram. As a result, the comparison between the experimental diffractograms and the simulated ones indicates a deformation of the crystal cell possibly due to the synthesis process in the presence of solid excipients [4].

The use of low dose rate during the acquisition of the HRTEM images enables to detect and study the extended structural defects in the pristine nanoparticles. This was done, for example, as shown in Figure 6, where a stacking fault, approximately in the middle of the nanoparticle, is detectable in the lattice fringes contrast, and it is reflected in the splitting of the relevant spots in the diffractogram on the right of the figure. In the diffractogram, the split concerns two couples of intensities corresponding to a spacing of (0.30 ± 0.01) nm, whereas the remaining couple of diffracted beams are due to a spacing of (0.32 ± 0.01) nm. The presence of extended defects in the structure of crystalline vincamine is related to the mechanochemistry process used for its synthesis, and it is at the origin of the broadening of the peaks in the relevant XRD measurements [4].

### 3.2. Caffeine/Glutaric Acid Co-Crystals

The in-line holography-based approach was applied here to the study of co-crystallization in nanoparticles, used in the design of a supramolecular structure with desired functional properties. Indeed, a co-crystal is a solid having two or three different molecules in the crystal structure and, therefore, it is particularly attractive for the application in engineering of composition of pharmaceutical phases [53]. For example, a molecule active against a particular disease can be associated in the same crystal cell of another drug, which is capable to reach a particular target or it is capable to overcome the cell barriers. On the other hand, co-crystals can easily exhibit polymorphism that can have deep influence on the properties of a drug, as demonstrated by the case of the anti-HIV drug ritonavir [54]. Single-particle studies by TEM in-line holography-based atomic resolution imaging have the possibility to access the crystal properties of individual nanoparticle of the drug, revealing the polytype and the influence of the synthesis process in controlling polymorphism phenomena [5]. The mechanochemical co-crystallization reaction in polymer-assisted grinding represents a well-controlled approach to the co-crystallization process, but it needs an appropriate understanding of the influence of polymer structure and polarity, together with the grinding conditions, on the synthesis results and, therefore, on the polytypes synthetized [55]. Caffeine (CAF) and glutaric acid (GLA) represent an ideal case study for the understanding of the co-crystallization process [5]; in particular, as solid excipient, ethylene glycol polymer chains of variable length and polarity were used. An in-line hologram of caffeine-glutaric acid (CAF-GLA) co-crystals is shown in Figure 7. TEM specimens were prepared dispersing the pristine powders on a copper grid previously covered by an amorphous carbon film, avoiding any pre-dispersion in liquid to prevent their possible modification. The aim is to have a low density of pristine nanoparticles on the copper grid to avoid accidental modification of the particle structure in the area illuminated by the electron beam, but not in the field of view of the microscopist and not under his direct monitoring.

The in-line hologram in Figure 7 is acquired in diffraction mode and shows few nanoparticles of ~10 nm in diameter, with high contrast in a field of view of ~500 nm. The density of electrons in the in-line hologram is ~1.2 e^−^Å^−2^. The low density of particles, on one hand, reduces the possibility of artifacts but, on the other hand, requires frequent and relatively wide movements of the specimen holder, with relevant specimen drift, to locate the particles and to put them properly in the field of view. It is therefore important to have an extremely low-dose approach, like the in-line hologram with defocused illumination, to check also the specimen drift until it stops, before switching the electron optical conditions to an intrinsically higher dose mode in the direct space HRTEM imaging, for an appropriate low dose exposure time. Figure 8 summarizes the atomic resolution information gained from one particle.

The particle in Figure 8 was synthetized by using a chain of polyethylene glycol of 1000 monomers as solid excipient. Figure 8c shows the HRTEM zoom on a nanoparticle oriented along a high symmetry zone axis, whose diffractogram in the area pointed by the arrow is shown in Figure 8b. The identification of the polytype can be univocally performed by comparing the experimental results with the simulations performed by using the Crystallographic Information File (CIF) available for the different polytypes of CAF-GLA system in the crystallography open database. The simulations here were performed by JEMS [56]. In particular, the particle in Figure 8 is a polytype I CAF-GLA oriented along the [−2, 0, 1] zone axis. The same approach was applied to the particle in Figure 9.

In the latter case, the particle belongs to the polytype II of CAF-GLA system and the relevant diffractogram shows that the particle is oriented along the [2, −4, 3] zone axis with respect to the primary electron beam. These kinds of experiments enable the study of the crystallography and the morphologic properties of individual pristine particles addressing the role of the synthesis conditions on the structure and the properties of the CAF-GLA co-crystals [5]. This approach enables the application of well-known and powerful electron microscopy methods, developed in materials science for materials robust to the radiation damage, to radiation-sensitive single particles.

### 3.3. Creatinine-Ferrihydrite Nanoparticles

In this last example of applications, the in-line holography-based atomic resolution imaging approach was applied to biologic traces of creatinine bound to ferrihydrate, which can be present in the bloodstream of patients suffering from acute kidney disease [57]. The pristine particles were placed on a copper grid and inserted in the high vacuum of the TEM specimen chamber without any pretreatment, like staining or similar procedures, usually employed on biologic specimens to increase the image contrast, or coating with carbon or metals, to prevent the charging effect and to partially protect the specimen from the electron irradiation. The experiments were performed at room temperature at an acceleration voltage of 200 kV. Figure 10a shows the in-line hologram as acquired, in diffraction mode, from a group of particles. The dark part on the left of Figure 10a is due to the mesh of the copper grid. The density of electron current is ~0.5 e^−^Å^−2^s^−1^, and the illuminated area is ~3.5 micron in diameter. Note that most of the particles visible in the in-line hologram show the evidence of some substructures. The origin of these substructures can be immediately recognized from the low magnification HRTEM image in Figure 10b, acquired on one of the particles visible in Figure 10a, where dark small particles appear embedded within the big one. The morphology, the size, and the contrast of the particle in Figure 10b are similar to those of globular protein of ferritin [58], but a further investigation rules this interpretation out.

As matter of the fact, the structure of the small dark particles, as measured from all the diffractograms, is compatible with the ferrihydrite of the ferritin, as shown, as an example, in Figure 11. In this case, the experimental data and the relevant simulation, performed by considering the known crystal structure of ferrihydrite [59], enable to index the particle as ferrihydrate oriented along the [4, 2, 1] zone axis with respect to the primary electron beam. In the figure are also reported the relevant Miller’s indexes together with their spacing. Nevertheless, all of the experimental data, HRTEM images, and relevant diffractograms acquired in the big particles away from the dark particles never reproduce what is simulated by using the known crystal structure of the globular protein of ferritin. Furthermore, we caution the reader that in all of our experiments we never observed the ferrihydrite fingerprint spacing at 0.25 nm. The discrepancy in the experimental data with respect to the hypothesis suggested by the morphologies and some structural data of the ferrihydrite can be understood in the light of what was reported in some studies of the interaction between ferrihydrite nanoparticles and creatinine and urea [56]. Note that the interaction and bonding between ferrihydrite and creatinine, or urea, is not likely to happen in the blood of a healthy organism, but could occur when some pathologic events, for example, rhabdomyolysis, determine the occurrence of the interaction of the content of the muscle cells, like ferritin, with waste substances, like urea and creatinine, contained in the blood stream. Indeed, rhabdomyolysis is a serious syndrome due to a direct or indirect muscle injury, and it results from the death of muscle fibers and release of their contents into the bloodstream. This can lead to serious complications, such as fatal renal failure, as the kidneys cannot remove waste and concentrated urine [60].

The studies by X-ray diffraction pattern reported in literature on the interactions between iron oxide nanoparticles and creatinine and urea show that the iron oxide fingerprint spacing at 0.25 nm is always absent when the nanoparticles of iron oxide are bound to creatinine or urea [56].

This evidence suggested the comparison of the experimental TEM results of the crystalline structure of the nanoparticle, like the one shown in Figure 10b, away from the iron oxide dark particles (Figure 11a,b), with the simulation performed starting from the known structure of creatinine and urea. The results of these comparisons show that in all the HRTEM experimental data collected, the relevant Fourier transforms are compatible with the structure of creatinine. An example is shown in Figure 12. Figure 12a is a HRTEM image focused on an area of the big envelope particle. Note the darker contrast due to the ferrihydrite nanoparticles and the lighter contrast due to the structure of the big particle embedding the ferrihydrite nanoparticles. Figure 12b shows the lattice contrast of the HRTEM image zoomed in the region marked by the blue square in Figure 12a. The relevant Fourier transform is shown in Figure 12c. The latter is compared with the pattern simulated starting from the known crystallographic information file (CIF) of creatinine structure shown in Figure 12d. In the latter, there are also reported the relevant Miller’s indexes of the crystal planes together with the spacing.

As a result of the TEM experiments and simulations, the particles present in the analyzed biological traces are therefore due to creatinine bound to ferrihydrate nanoparticles. This result helps understand that the biological traces studied by TEM could be due to the serum of an organism with a pathologic bonding of creatinine to ferrihydrite due to rhabdomyolysis. The detailed experimental data on the biologic specimen, collected at room temperature and by using the pristine material, were successfully obtained due to the capability of the in-line holography low dose approach to study a specimen by high-energy electrons despite its sensitivity to the radiation damage. The approach achieves high spatial resolution information of individual biologic particles as whole, and within their inner structure, enabling to detect eventual anisotropy within their volume.

## 4. Conclusions and Future Perspectives

Atomic resolution single-particle TEM studies of radiation-sensitive organic and inorganic matter are essential for the complete understanding of the properties of this important class of materials and therefore for the advances in biology, pharmacology, medicine, material science, physics, etc. Accurate true imaging of single particles is also a prerequisite for a successful and more reliable structural modeling by cryo-EM, providing a priori information that drives the modeling. Unfortunately, the sensitivity to radiation damage prevents a straightforward use of the powerful TEM/STEM atomic resolution methods developed for materials robust to radiation as not only the particle irradiation for the imaging acquisition itself, but all the steps necessary for a meaningful quantitative single particle imaging experiment can destroy, or at least damage, the case of interest.

Here, it was established how in-line holography coupled with HRTEM enables the performing of extremely low dose experiments on radiation-sensitive nanoparticles of organic and inorganic materials, thus accessing the properties of single particles allowing the understanding of their structural properties and enabling the correlation to their performances. All of these experiments were not on specially designed specimens, but on standard specimens of pristine nanoparticles with different structure, chemistry, and morphology. This true single-particle study of radiation-sensitive matter opens new perspective in a variety of scientific disciplines. In particular, here the use of in-line holography allows setting up the electron optics to provide reliable low dose and low dose rate atomic resolution imaging, to find the particle of interest accessing immediately its crystalline status and shape, to monitor the eventual specimen drift, and to check when it stops to make atomic resolution imaging possible. All these steps delivering a safe density of current between 0.1 to 10 e^−^Å^−2^s^−1^ and monitoring the eventual structural damage. The experiments on nanoparticles of drugs and of organic materials have shown that the pristine properties of single particles of radiation-sensitive matter can be studied at atomic resolution even at room temperature by using electrons of 200 keV.

The results shown so far regard the combination of in-line holography and low dose rate HRTEM to provide true atomic resolution imaging on single particle. Moreover, the recent understanding on in-line holography and digital reconstruction of in-line holograms [44,45,61] indicates a further advancement in the application of in-line holography to provide, by itself, detailed information of the atomic structure of organic molecules while delivering low dose of electrons to the specimen. These features together with the successful theoretical demonstration that tridimensional structure of organic molecules can be recovered by in-line holography from a single projection [52] pave the way to a new direct knowledge of the atomic structure of organic and inorganic materials by electrons in a TEM.

## Figures and Tables

**Figure 1 materials-13-01413-f001:**
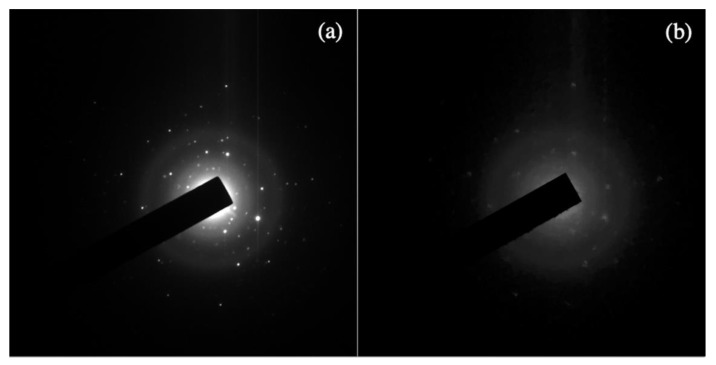
Selected area electron diffraction patterns of a specimen consisting of nanoparticles of Vincamine; (**a**) after an exposure to the electron beam with a density of current of 300 e^−^Å^−2^s^−1^ for 0.01 sec [4] and (**b**) after an exposure to the same current density for 0.3 s.

**Figure 2 materials-13-01413-f002:**
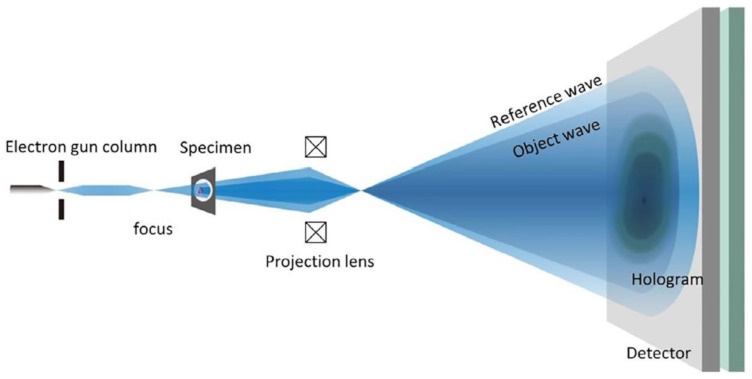
Scheme of the in-line hologram as firstly proposed by Dennis Gabor.

**Figure 3 materials-13-01413-f003:**
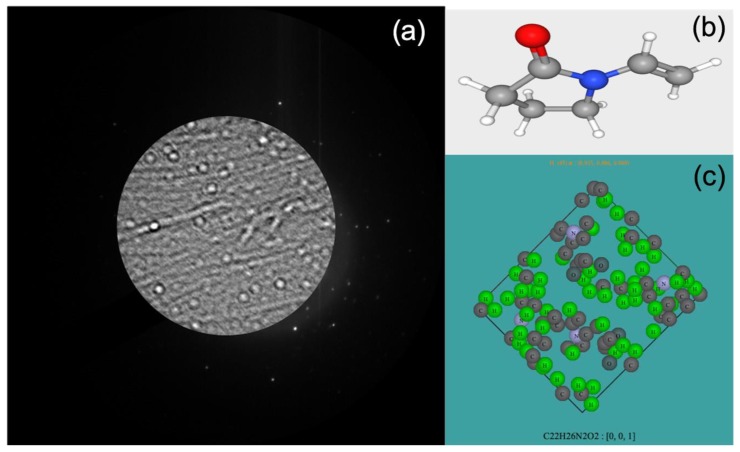
(**a**) In line-hologram on Vinpocetine and polyvinylpyrrolidone acquired in diffraction mode. (**b**) 3D Chemical structure of polyvinylpyrrolidone (gray atoms: C; white atoms: H; red atoms: O; blue atoms: N). (**c**) Crystal cell of Vinpocetine in [0,0,1] zone axis.

**Figure 4 materials-13-01413-f004:**
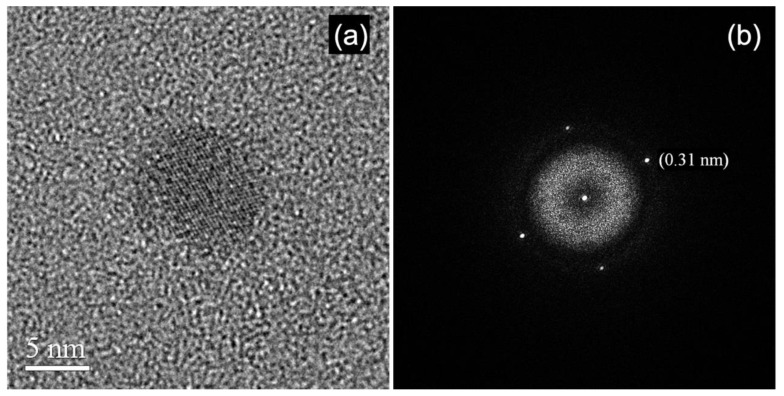
Low dose rate HRTEM image (**a**) and relevant diffractogram from a crystalline particle of Vinpocetine (**b**). The lattice spacing measured on the diffractogram is reported.

**Figure 5 materials-13-01413-f005:**
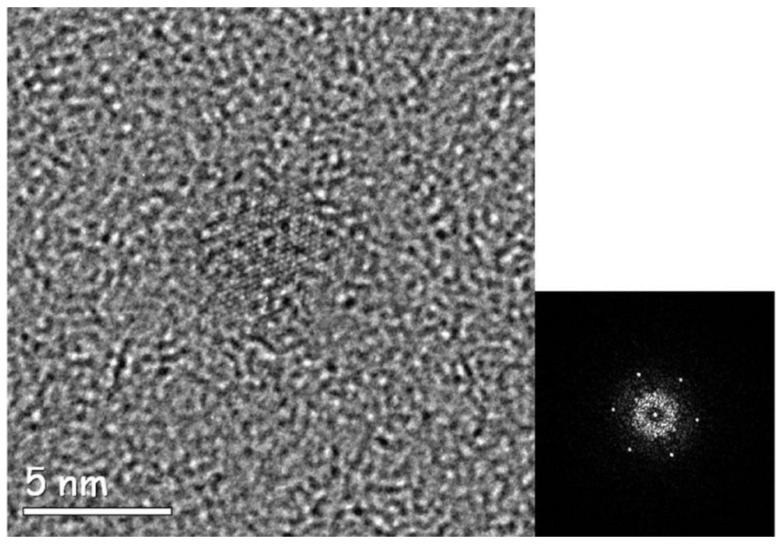
Left: Low dose rate HRTEM of a particle of vincamine. Right: The relevant high symmetry diffractogram (reprinted by courtesy from Hasa et al. [4]).

**Figure 6 materials-13-01413-f006:**
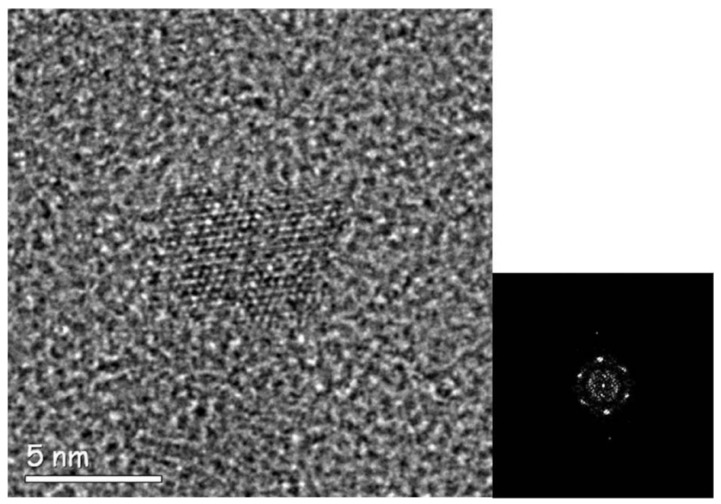
Left: Low dose rate HRTEM of a particle of vincamine. Right: The relevant high symmetry diffractogram. The lattice fringes in the central part of the particle and the spot splitting in the diffractogram point the presence of an extended structural defect in the crystal structure (reprinted by courtesy from Hasa et al. [4]).

**Figure 7 materials-13-01413-f007:**
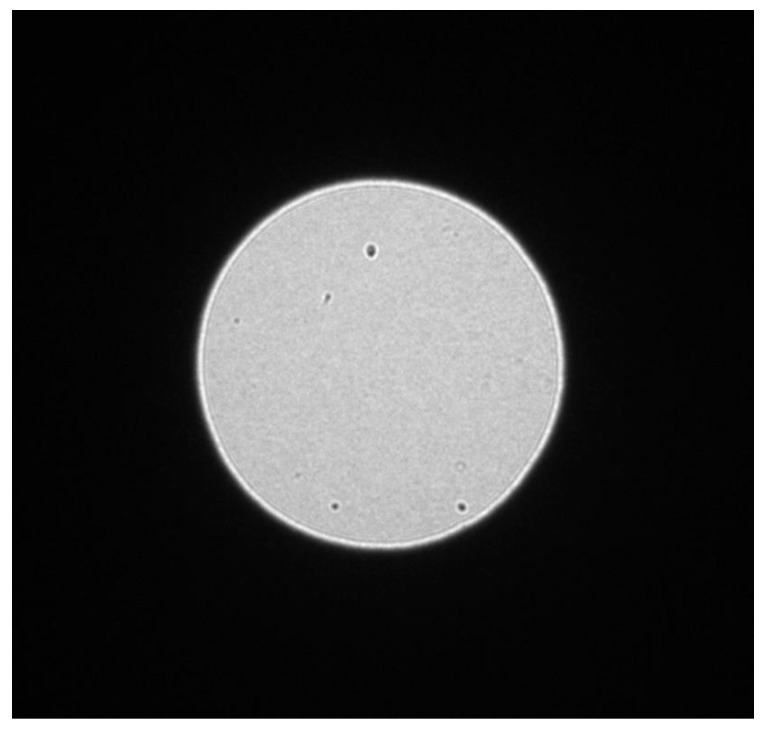
In line-hologram on cocrystals of caffeine and glutaric acid as acquired in diffraction mode and exposed to 1.2 e^−^ Å^−2^.

**Figure 8 materials-13-01413-f008:**
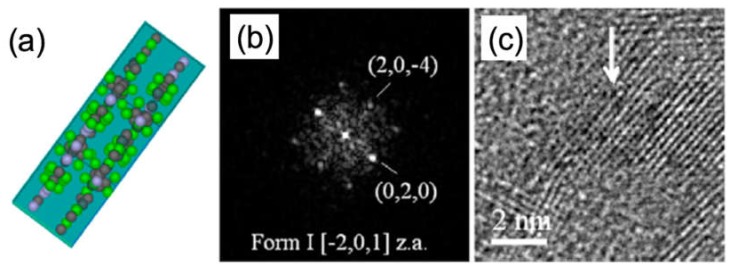
Caffeine-glutaric acid (CAF-GLA) co-crystal of polytype I [5]. The kind of polytype of the nanoparticle is univocally determined by comparing the structure of type I, viewed along the [−2, 0, 1] zone axis in (**a**), with the experimental diffractogram (**b**) and the HRTEM image (**c**). The arrow points the region from which the diffractogram was extracted (reprinted by courtesy from Hasa et al. [5]).

**Figure 9 materials-13-01413-f009:**
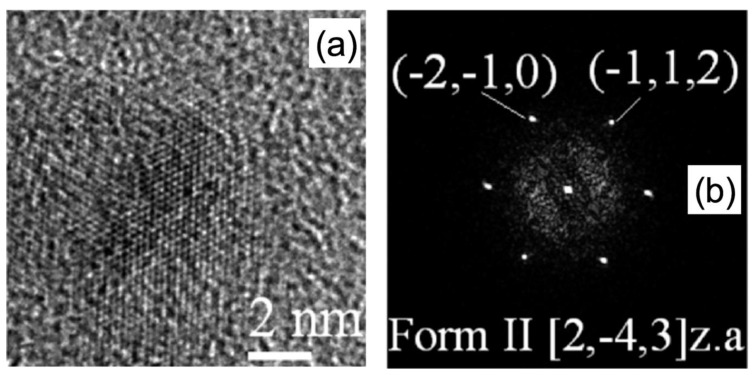
(**a**) HRTEM image of a caffeine (CAF) glutaric acid (GLA) polytype II [5] particle along with (**b**) the relevant diffractogram (reprinted by courtesy from Hasa et al. [5]).

**Figure 10 materials-13-01413-f010:**
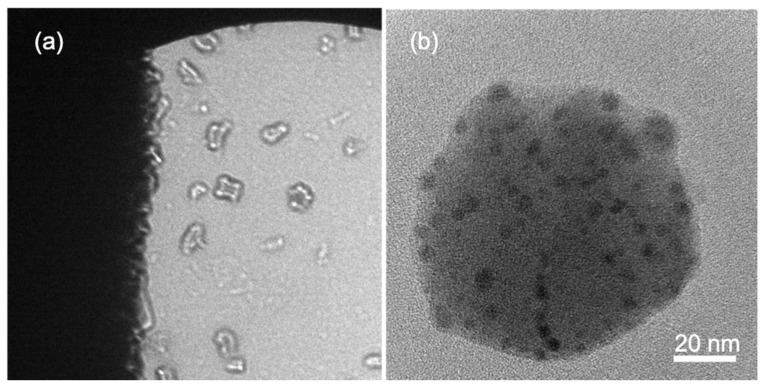
(**a**) In-line hologram of biologic particles. (**b**) Low-magnification HRTEM of one of the biologic nanoparticles; note the dark smaller nanoparticles within the big one.

**Figure 11 materials-13-01413-f011:**
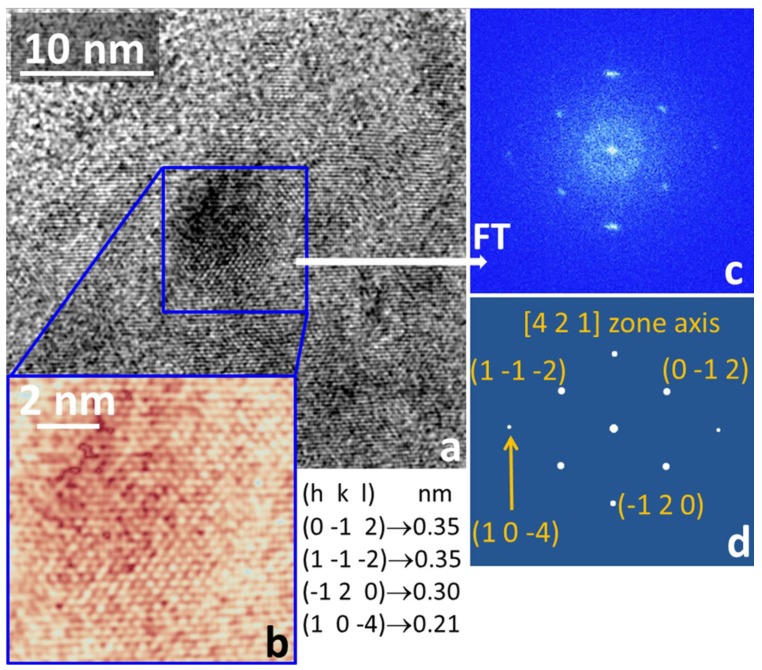
Representative results on the small dark particles. (**a**) HRTEM image focused on one of the small and dark nanoparticles shown in Figure 10; (**b**) zoom inside the blue square of panel (**a**) (false color output); (**c**) Fourier transform relevant to the zoomed area in panel (**b**); (**d**) simulation, with the Miller indexes associated to some spots and the corresponding lattice spacing. Simulations show that the pattern of panel (**c**) is compatible with the ferrihydrate in the [4, 2, 1] zone axis.

**Figure 12 materials-13-01413-f012:**
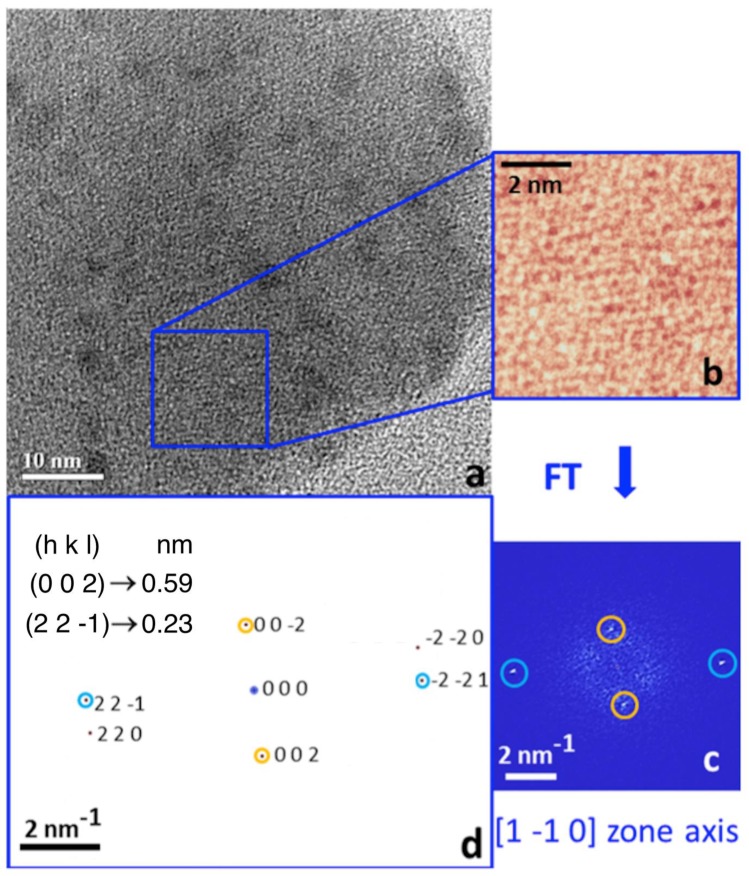
(**a**) Characterization of creatinine particles; HRTEM image; (**b**) magnified view of the square region marked in (**a**); (**c**) Fourier transform of (**b**); (**d**): simulation of the diffraction pattern of creatinine in [1, −1, 0] zone axis with reported the lattice spacing relevant to the observed intensities. The yellow and pale-blue circles in the diffraction pattern simulation mark the correspondence with the circles around the experimental spots in panel (**c**).

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
