# Peer review of "In-Line Holography in Transmission Electron Microscopy for the Atomic Resolution Imaging of Single Particle of Radiation-Sensitive Matter"

_materials, 2020, doi:10.3390/ma13061413_

Round 1

Reviewer 1 Report

The submission is certainly interesting, but some points should be worked on before publication:

  1. The reader who is not familiar with the materials presented is wondering how the crystal structure of vinpocetine look like. Is there a model that the author could show?
  2. It is not made clear enough how it comes that we see an image of the sample in diffraction mode.
  3. In Figs 11c  & 12c the author writes "diffractogram", while the images are marked "FT", i.e. a fouries transform image. This is not a diffractogram, only a virtual one. This should be clarified.
  4. The lay-out of the references is not correct; web-citations should contain an access date.
  5. The unit of the density of current is given in two different modes, only one of which is correct: the one in the abstract.
  6. The presentation of the equations is lousy. Additionally, they do not exactly belong into the section "methods".
  7. There are some English mistakes and some typos.

Author Response

Answers to referee 1:

Comments and Suggestions for Authors

R: The submission is certainly interesting, but some points should be worked

on before publication:

A: As first I would like to thank the reviewers 1 for his comments and suggestions to improve the paper. In the following the answer to each point raised by the reviewer1:

1.R: The reader who is not familiar with the materials presented is wondering

how the crystal structure of vinpocetine look like. Is there a model that

the author could show?

1.A: I added the model for the vinpocetine and in particular its crystal cell in (0,0,1) projection. I also added the 3D chemical structure of PVP. To this aim I provided a new figure 3 where this information was added. I modified accordingly the relevant caption.

2.R: It is not made clear enough how it comes that we see an image of the sample in diffraction mode.

2.A: When the electron probe is focused above or below the specimen plane, each diffracted disc in the reciprocal plane contains a shadow image of the direct plane of the specimen. The magnification of this shadow image is related to distance “u” between the focal plane and the specimen and on the distance “v” between the specimen plane and the plane of view. From the geometric optic (see the references in the paper) M=v/u. I added a relevant statement in the emended version of the paper and a new reference (#46 in the emended version).

3.R: In Figs 11c & 12c the author writes "diffractogram", while the images are

marked "FT", i.e. a fouries transform image. This is not a diffractogram,

only a virtual one. This should be clarified.

3.A: The statement has been emended and the corrections are marked in yellow, for the reader convenience, in the marked emended version of the paper

4.R: The lay-out of the references is not correct; web-citations should contain

an access date.

  1. A: the references layout has been checked and access date for web citation added
  1. R: The unit of the density of current is given in two different modes, only

one of which is correct: the one in the abstract.

  1. A: I made the notation homogeneous
  1. R: The presentation of the equations is lousy. Additionally, they do not

exactly belong into the section "methods".

  1. A: The style of the equations has been changed.
  1. R: There are some English mistakes and some typos.
  1. A: English mistakes and typos have been corrected.

Reviewer 2 Report

The article entitled as In-line holography in transmission electron microscopy for the atomic resolution imaging of single particle of radiation sensitive matter brings out some information (but not enought) on the how the in-line holography helps to carry out HRTEM of radiation sensitive materials just published but few novel information about it.

There are three sections in Results and discussion. The first one Vincamine were published elsewhere but doesn’t show a in line-hologram to localize the particle or how the in-line holography strengthen the HRTEM study that I think is the object of the manuscript. The second one Caffeine/glutaric acid co-crystals show the in line-hologram but the HRTEM study was published years ago. The third one Creatinine-ferrihydrite nanoparticles shows the new information of in-line holography and HRTEM study of this kind of nanoparticles. The author should show in more detail how the in-line holography helps the HRTEM study and the comparison with another methodologies. Moreover, the author should address the following shortcoming:

  • The electron dose flux was calibrated using a Faraday cup or another instrument?
  • Are there are other strategies that Low-dose technique or cool specimen to acquire HRTEM studies of sensitive nanoparticles?
  • Is it really the first time that in-line holography enables the HRTEM studies of nanoparticles?
  • The in-line holography images of the manuscript show a low magnification image. In which way the In-line holography help to estimate and make it as close as possible to Scherzer defocus?

Minor changes should be addressed:

1) Equations look bad.

2) Unify “fig.”, “Fig.” or “figure”.

3) Always leave a space between the value and the units.

4) The format style of the reference should be corrected.

Author Response

Answers to referee 2:

Comments and Suggestions for Authors

R: The article entitled as In-line holography in transmission electron microscopy for the atomic resolution imaging of single particle of radiation sensitive matter brings out

some information (but not enought) on the how the in-line holography helps to carry out HRTEM of radiation sensitive materials just published but few novel information

about it.

A: I would like to thank the reviewer 2 for his/her comments and suggestions. The aim of the paper is to show how the use of in line hologram enables atomic resolution imaging on single nanoparticles of radiation sensitive specimen. Since the first introduction of the method, about eight years ago, successful experiments have been carried out and published in the literature (ref 4-5 in the paper) but a full description of the technique has not been reported yet. To our knowledge, there are no other way available to study pristine radiation sensitive matter constitutes by single nanoparticles on standard specimen, even by using the most advanced equipment now available. Some experiments made in the last weeks in our groups further confirmed this aspect.

R: There are three sections in Results and discussion. The first one Vincamine were published elsewhere but doesn’t show a in line-hologram to localize the particle or how the in-line holography strengthen the HRTEM study that I think is the object of the manuscript. The second one Caffeine/glutaric acid co-crystals show the in line-hologram but the HRTEM study was published years ago. The third one Creatinine-ferrihydrite nanoparticles shows

the new information of in-line holography and HRTEM study of this kind of nanoparticles. The author should show in more detail how the in-line holography helps the HRTEM study and the comparison with another methodologies.

A: As described in the paper, there is no other way to survey the specimen to find representative particles, to adjust the electron optical conditions and to check the specimen drift without destroying the specimen itself. To our knowledge this is the only method available on this kind of specimen and this is the reason way, to our knowledge, there were no atomic resolution imaging studies on this kind of samples before the use of this approach. All these aspects, related to the methodology and to its practical application to the study of radiation sensitive materials, which is the aim of the paper, are described in detail along the manuscript; some cases study are reported as examples.

R: The electron dose flux was calibrated using a Faraday cup or another instrument?

A: The current was measured by Faraday’s cup. A statemen was added in the amended paper at row 318.

R: Are there are other strategies that Lowdose technique or cool specimen to acquire

HRTEM studies of sensitive nanoparticles?

A: There are other standard low-dose approaches, as reported in the paper, but these were not effective in the reported examples to provide reliable information on the pristine particles. The imaged particles were always damaged by using standard approaches and without the hologram based survey the microscopist gropes in the dark. All these aspects were discussed in the paper.

R: Is it really the first time that in-line holography enables the HRTEM studies of

nanoparticles?

A: As reported in the paper and in the references therein, for example from the row 248 to 258, in-line holography was used plenty of times for a variety of application in electron microscopy. To our knowledge this is the first time that it is used according to the procedures here described to study successfully pristine radiation sensitive nanoparticles at atomic resolution.

R: The in-line holography images of the manuscript show a low magnification

image. In which way the In-line holography help to estimate and make it as close as

possible to Scherzer defocus?

A: As described in the paper, for example from row 298 to row 304, the hologram is acquired in the reciprocal space after that in the direct space the electron optical conditions are adjusted for HRTEM imaging. Then the other tunings are made all in the reciprocal space and when all the parameters are suitable for low-dose, low-dose rate HRTEM acquisition, the detector is conjugated to the direct space for the image acquisition where the optical conditions were not changed. Note that, operating in the reciprocal space enables an accurate tuning of the convergence angle and on the illuminated area.

Minor changes should be addressed:

R: Equations look bad.

A: The equation style was emended

R: Unify “fig.”, “Fig.” or “figure”.

A: This has been done in the emended paper

R: Always leave a space between the value

and the units.

A: The paper has been emended accordingly

R: The format style of the reference should

be corrected.

A: The paper has been checked

Round 2

Reviewer 2 Report

The author has responded satisfactorily to the comments.